# Luteolin Treatment Ameliorates Brain Development and Behavioral Performance in a Mouse Model of CDKL5 Deficiency Disorder

**DOI:** 10.3390/ijms23158719

**Published:** 2022-08-05

**Authors:** Marianna Tassinari, Nicola Mottolese, Giuseppe Galvani, Domenico Ferrara, Laura Gennaccaro, Manuela Loi, Giorgio Medici, Giulia Candini, Roberto Rimondini, Elisabetta Ciani, Stefania Trazzi

**Affiliations:** 1Department of Biomedical and Neuromotor Sciences, University of Bologna, 40126 Bologna, Italy; 2Department of Medical and Surgical Sciences, University of Bologna, 40126 Bologna, Italy

**Keywords:** CDKL5 deficiency disorder, luteolin, microglia, dendritic spines, neurogenesis, neuronal plasticity

## Abstract

CDKL5 deficiency disorder (CDD), a rare and severe neurodevelopmental disease caused by mutations in the X-linked *CDKL*5 gene, is characterized by early-onset epilepsy, intellectual disability, and autistic features. Although pharmacotherapy has shown promise in the CDD mouse model, safe and effective clinical treatments are still far off. Recently, we found increased microglial activation in the brain of a mouse model of CDD, the *Cdkl*5 KO mouse, suggesting that a neuroinflammatory state, known to be involved in brain maturation and neuronal dysfunctions, may contribute to the pathophysiology of CDD. The present study aims to evaluate the possible beneficial effect of treatment with luteolin, a natural flavonoid known to have anti-inflammatory and neuroprotective activities, on brain development and behavior in a heterozygous *Cdkl5* (+/−) female mouse, the mouse model of CDD that best resembles the genetic clinical condition. We found that inhibition of neuroinflammation by chronic luteolin treatment ameliorates motor stereotypies, hyperactive profile and memory ability in *Cdkl5* +/− mice. Luteolin treatment also increases hippocampal neurogenesis and improves dendritic spine maturation and dendritic arborization of hippocampal and cortical neurons. These findings show that microglia overactivation exerts a harmful action in the *Cdkl5* +/− brain, suggesting that treatments aimed at counteracting the neuroinflammatory process should be considered as a promising adjuvant therapy for CDD.

## 1. Introduction

CDKL5 deficiency disorder (CDD), a severe neurodevelopmental disorder caused by mutations in the X-linked *CDKL*5 gene [1,2], is characterized by a complex symptomatology, including early infantile onset refractory epilepsy, hypotonia, and severe cognitive, motor, visual, and sleep disturbances [3]. The majority of patients are heterozygous females, with fewer hemizygous males reported [4,5]. During the past few years, the role of CDKL5 in brain development and function has been elucidated using an animal model of CDD, the *Cdkl*5 knockout (KO) mouse [6,7,8,9]. *Cdkl*5 KO mice recapitulate different features of CDD, exhibiting autistic-like behavior, severe impairment in learning and memory, altered social interactions, visual and respiratory deficits, and motor stereotypies [6,7,8,9,10,11,12,13]. These defects are associated with neuroanatomical alterations; reduced neuronal branching, spine maturation, and connectivity have been observed in the cortex [14,15,16,17,18,19] and in the hippocampal region [7,10,11,20,21,22] of *Cdkl*5 KO mice. In addition, *Cdkl*5 KO mice are characterized by an increased rate of apoptotic cell death in the hippocampal dentate gyrus and by increased susceptibility to neurotoxic/excitotoxic stress of pyramidal hippocampal neurons [10,23,24], indicating that an absence of Cdkl5 increases neuronal vulnerability.

Increasing evidence indicates that marked microglial activation, leading to neuroinflammation, is important in pathological neurodevelopmental contexts, including autism spectrum disorders [25,26], obsessive compulsive disorder [27,28], schizophrenia [29,30], Tourette’s syndrome [31], Fragile X syndrome [32,33], Rett syndrome [34,35,36], and Down syndrome [37]. Interestingly, in addition to the traditional macrophage-type role and source of multiple neurotoxic factors, microglia have more recently been revealed as critical players in brain development and cognitive functions [38]. It is conceivable that microglia could engage in crosstalk with neurons and release neurotrophins [39] or other signaling factors that are able to regulate synaptic maturation and transmission, thereby modulating synaptic homeostasis. Microglia may also sense signals from the surrounding environment and have regulatory effects on neurogenesis [40]. Importantly, several preclinical findings demonstrated that inhibition of neuroinflammation can reduce the severity of several neurodevelopmental disorders, such as Down syndrome [41,42], Rett syndrome [43], neurodevelopmental disorders derived from maternal immune activation [44], cerebral palsy [45], autism spectrum disorder [46,47], and schizophrenia [48]. 

Natural flavonoids, such as luteolin, exert interesting pharmacological effects with their antioxidative, anti-inflammatory properties coupled with their capacity to modulate key cellular enzyme functions [49]. The flavonoid molecular structure confers the capability of reacting with and neutralizing reactive oxygen species (ROS), thus acting as a scavenger of free radicals. Moreover, it has been demonstrated that luteolin downregulates IL-1β, IL-6, and TNF-α, and directly counteracts NF-kB, MAPK, and JAK/STAT inflammatory pathways by reducing the inflammatory status in cellular models of inflammation [50,51], suggesting its potential usefulness as a natural treatment option to counteract neuroinflammation.

Recently, we found a microglia overactivation status in the brain of *Cdkl*5 KO mice [52]. Interestingly, treatment with luteolin, by restoring microglia alterations, counteracts hippocampal neuron cell death, and rescues NMDA-induced excitotoxic damage in *Cdkl*5 KO mice [52], suggesting the beneficial effect of microglia activity restoration on Cdkl5-null neuron survival. Despite the well-documented evidence of the involvement of neuroinflammatory processes in the pathophysiology of neurodevelopmental disorders [53,54,55,56,57], it is not known whether CDKL5-related microglial overactivation plays a role in dendritic pathology and, consequently, in behavioral abnormalities in CDD. 

In this study, we provide novel evidence that luteolin treatment, by restoring microglia alterations and transiently boosting BDNF/TrkB signaling, improved or even restored hippocampal neurogenesis, dendritic length and branching of hippocampal and cortical pyramidal neurons, and spine maturation in *Cdkl*5 KO mice. Notably, neuroanatomical changes induced by the anti-inflammatory treatment with luteolin were associated with an improvement in behavioral performance in *Cdkl*5 KO mice, strengthening the prospect that CDD patients could benefit from anti-inflammatory treatments.

## 2. Results

### 2.1. Treatment with Luteolin Ameliorates Behavioral Deficits in Cdkl5 +/− Mice

In a previous study, we found that treatment with luteolin restored microglia overactivation in *Cdkl*5 KO mice [52]. In the current study, we were interested in establishing whether microglia overactivation contributes to behavioral impairment in *Cdkl*5 KO mice. To this purpose, we treated three-month-old heterozygous female (+/−) *Cdkl*5 KO mice with luteolin or a vehicle for 20 days. In order to investigate treatment effects on impaired locomotor activity, stereotypic and autistic-like behaviors, and memory, a battery of behavioral tests was carried out, starting from the eighth day of treatment and lasted twelve days, during which treatment was not interrupted (Figure 1A). 

No differences were found in the body weight of the experimental subjects (Appendix A), indicating that animal well-being was not affected by chronic luteolin treatment.

We first evaluated the behavioral effect of luteolin by assessing motor stereotypies through a hind-limb clasping test. *Cdkl*5 +/− mice spent more time during the test session in the clasping position, in comparison with wild-type (*Cdkl*5 +/+) mice (Figure 1B). Luteolin treatment led to a significant reduction in clasping time in *Cdkl*5 +/− mice (Figure 1B), indicating a treatment-induced improvement of motor stereotypies. 

*Cdkl*5 +/− mice also showed a hyperactive profile when exposed to a new environment, as demonstrated by the stereotyped behavior (repetitive vertical jumping) and the higher distance travelled and increased velocity, compared to *Cdkl*5 +/+ mice, during the open field task (Figure 1C,D). Interestingly, treatment with luteolin normalized the stereotyped behavior in *Cdkl*5 +/− mice (Figure 1C) and improved the hyperactive phenotype (Figure 1D). Analysis of the time spent in the center of the arena did not reveal any differences between the groups concerning the anxiety-like behavior in this task (Appendix A). 

Finally, we performed a passive avoidance (PA) test to evaluate the functional effect of the treatment on the memory ability of *Cdkl*5 +/− mice. While during the first day of the test all groups showed similar step-through latencies (Figure 1E), during the second day trial, *Cdkl*5 +/− mice were severely impaired at performing this task, showing a reduced latency to enter the dark compartment compared to control mice (Figure 1E). Albeit not statistically significant, treatment with luteolin increased the latency with which *Cdkl*5 +/− mice entered the dark compartment (Figure 1E), indicating an improvement in memory performance. 

To confirm the anti-inflammatory effect of luteolin in the Cdkl5-null brain [52], we evaluated, as an index of microglial activation, the body size and roundness of microglial cells in the cortex and hippocampus of *Cdkl*5 +/− mice. A twenty-day treatment with luteolin fully recovered these microglia alterations (Appendix A) both in the hippocampus and somatosensory cortex, suggesting that inhibition of microglia overactivation underlies, at least in part, the behavioral deficits in *Cdkl*5 KO mice.

### 2.2. Treatment with Luteolin Promotes Neurogenesis in the Hippocampus of Cdkl5 +/− Mice 

Recent evidence has shown that luteolin promotes hippocampal neurogenesis in a mouse model of Down syndrome [41]. To establish whether luteolin treatment increases proliferation rate and promotes neurogenesis of newborn hippocampal neurons in *Cdkl*5 +/− mice, we evaluated the number of Ki67-, BrdU-, and DCX-positive cells in the subgranular and granular zone of the hippocampal dentate gyrus (DG). 

We used Ki-67 immunohistochemistry to estimate the size of the population of actively dividing cells in the DG at the end of treatment with luteolin. We found more Ki-67-positive cells (Ki-67+) after treatment with luteolin in *Cdkl*5 +/− mice (Figure 2A,E), indicating that luteolin treatment increased the pool of proliferating precursors. In order to confirm the effect of treatment on hippocampal cell proliferation, mice were injected with BrdU at the end of the treatment (Figure 1A) and the number of BrdU-positive cells in the DG was evaluated 24 h after the injection. 

As expected, we found an increase in the number of BrdU-positive cells (BrdU+) in the DG of treated *Cdkl*5 +/− mice, confirming that luteolin treatment increased the proliferation rate of precursor granule cells of the DG (Figure 2B,E). 

In order to assess the effect of luteolin treatment on neuronal survival, the number of DCX-positive (DCX+) newborn granule cells was quantified. *Cdkl*5 +/− mice had a lower number of DCX+ cells compared to their wild-type counterparts (Figure 2C,E). After 20 days of luteolin treatment, not only was the number of newborn granule cells in *Cdkl*5 +/− mice recovered, but it was even significantly increased compared to the number of DCX+ cells present in wild-type mice (Figure 2C,E). To establish the impact of the increased neurogenesis in luteolin-treated *Cdkl*5 +/− mice on the overall granule cell number, we stereologically evaluated the granule cell density in the DG of vehicle and luteolin-treated *Cdkl*5 KO mice. As previously reported, the granule cell layer of *Cdkl5* +/− mice had a reduced granule cell density [10]. Consistent with the higher number of DCX+ cells in *Cdkl*5 +/− mice (Figure 2C), we noted a higher number of Hoechst-stained nuclei in the granule cell layer of luteolin-treated *Cdkl5* +/− mice, indicating that luteolin treatment restored cellularity in the DG of *Cdkl5* +/− mice (Figure 2D). 

### 2.3. Treatment with Luteolin Improves Maturation of Newborn Cells in the Dentate Gyrus of Cdkl5 +/− Mice

To establish the effect of inhibition of microglia overactivation on dendritic development of newborn granule cells, we examined the dendritic morphology of DCX+ cells (Figure 3). We examined the total length of the dendritic tree and the number and mean dendritic length of segments. As previously found [10,52], loss of Cdkl5 causes dendritic hypotrophy of newborn cells in the dentate gyrus of *Cdkl*5 +/− mice (Figure 3A,D). 

This defect was mainly due to a lower number of dendritic branching (Figure 3B,D). *Cdkl*5 +/− mice had fewer branches of the orders higher than 5, while a difference in mean length was present only in branches of order 9 (Figure 3E). Luteolin treatment recovered the total dendritic length of newborn granule cells in *Cdkl*5 +/− mice to wild-type levels (Figure 3A,D). The recovery was primarily due to an increase in the mean length of dendrites that became significantly longer than that of wild-type mice (Figure 3C,D), especially for the first 7 orders (Figure 3E). In contrast, treatment did not improve the number of branches of DCX+ cells in *Cdkl*5 +/− mice (Figure 3B,D).

### 2.4. Treatment with Luteolin Improves the Dendritic Architecture in Hippocampal and Cortical Neurons of Cdkl5 +/− Mice

Previous evidence showed that hippocampal and cortical pyramidal neurons of *Cdkl*5 KO mice are characterized by severe dendritic hypotrophy [7,8,10,11,17,18,22,58]. In order to establish whether a prolonged treatment with luteolin (20 days) also ameliorates dendritic arborization defects of neurons already present at birth, we examined the dendritic tree of Golgi-stained pyramidal neurons in the hippocampus and cortex of *Cdkl*5 +/− mice (Figure 4A). Both apical and basal dendrites of pyramidal hippocampal and cortical neurons of *Cdkl*5 +/− mice had a shorter total length (Figure 4B,C) and a reduced number of branches (Figure 4D,E), in comparison with wild-type (+/+) mice. In *Cdkl*5 +/− mice treated with luteolin, the total dendritic length and number of branches underwent an increase and became similar to those of the wild-type (+/+) mice (Figure 4B–E). 

We next examined the effect of treatment on each dendritic order separately. In hippocampal neurons of *Cdkl*5 +/− mice, dendritic hypotrophy was due to a reduced number of branches of an order higher than 3 (Figure 5A) and, in the basal dendrites, a lack of branches of an order higher than 8 (Figure 5A, black arrows). In cortical neurons, *Cdkl*5 +/− mice had a similar number of branches of orders 1 and 4 for apical dendrites and 1 and 2 for basal dendrites compared to wild-type (+/+) mice but fewer branches of subsequent orders (Figure 5B). Unlike wild-type (+/+) mice, *Cdkl*5 +/− mice lacked branches of orders 10 and 11 for apical dendrites and 7 and 8 for basal dendrites (Figure 5B, black arrows). In *Cdkl*5 +/− mice treated with luteolin, the number of branches became similar to the wild-type (+/+) mice (Figure 5A,B). Regarding the mean length of dendritic branches of individual orders, a reduction was found in the mean branch length of order 9 for apical dendrites and orders 6–7 for basal dendrites of hippocampal neurons (Appendix A), and of orders 7–8 for apical and 5–6 for basal dendrites of cortical neurons (Appendix A). Importantly, luteolin-treated *Cdkl*5 +/− mice underwent an increase in the mean length of branches of these orders (Appendix A). 

### 2.5. Treatment with Luteolin Improves Spine Maturation in the Brain of Cdkl5 +/− Mice 

Alterations in dendritic spine morphology and defects in synaptogenesis have been consistently reported in *Cdkl*5 KO mice [7,10,11,14,20,21,22]. To investigate the effect of luteolin treatment on dendritic spine development, we analyzed the dendritic spine morphology of Golgi-stained hippocampal and cortical pyramidal neurons. Separate counts of different classes of dendritic spines confirmed that pyramidal neurons of both hippocampal field CA1 (Figure 6A,B) and the somatosensory cortex (Figure 6C) of *Cdkl*5 +/− mice had a higher percentage of immature spines (filopodium-like, thin-shaped, and stubby-shaped) and a reduced percentage of mature spines (mushroom and cup shaped), compared to wild-type (+/+) mice. After luteolin treatment, *Cdkl*5 +/− mice showed a percentage of mature and immature spines that was similar to *Cdkl*5 +/+ mice (Figure 6A–C), suggesting that treatment was able to restore dendritic spine maturation. 

Since a proper dendritic spine morphology is essential for the formation of synaptic contacts, we further evaluated the number of immunoreactive puncta for PSD-95 (postsynaptic density protein 95) in the hippocampus of *Cdkl*5 +/− mice. *Cdkl*5 +/− mice displayed a strong reduction in the number of PSD-95-positive puncta compared to *Cdkl*5 +/+ mice (Figure 6D,E); luteolin treatment promoted a significant increase in the number of synaptic puncta in *Cdkl*5 +/− mice (Figure 6D,E).

### 2.6. Treatment with Luteolin Transiently Boosts BDNF/TrkB Signaling Pathways in the Cortex of Cdkl5 +/− Mice 

BDNF/TrkB signaling activates several intracellular pathways that play an important role in neuronal differentiation, dendritic morphogenesis, neuroprotection and modulation of synaptic interactions, which are critical for cognition and memory [59]. Among these, TrkB phosphorylation leads to phosphatidylinositol 3-kinase (PI3K)/Akt and extracellular signal-regulated kinase/Erk activation [60]. We had previously shown that a 7-day luteolin treatment promotes an increase in brain-derived neurotrophic factor (BDNF) levels in the cortex of *Cdkl*5 +/− mice ([52]; Appendix A). Surprisingly, we found that luteolin-dependent increased BDNF levels were not maintained after 20 days of treatment (Appendix A). To investigate the pathways activated by luteolin-induced BDNF expression and their maintenance over time, we first analyzed the levels of TrkB phosphorylation using Western blot in cortical homogenates of *Cdkl*5 +/− mice. As previously reported in male *Cdkl*5 -/Y mice [17], although no significant differences in BDNF levels were observed (Appendix A), a significantly lower level of P-TrkB was present in the *Cdkl*5 +/− cortex compared to that of wild-types (+/+) (Figure 7B,C). Importantly, a 7-day luteolin treatment normalized P-TrkB levels to those of the wild-type condition by increasing BDNF levels (Figure 7A). As expected, after 20 days of luteolin treatment, the P-TrkB levels were no longer different from those of untreated *Cdkl*5 +/− mice (Figure 7A). We then examined the main downstream effectors of the TrkB pathway. Predictably, we found a significantly increased Erk and Akt phosphorylation in the cortex of 7-day luteolin treated *Cdkl*5 +/− mice, in comparison with wild-type (+/+) mice (Figure 7B,C), whereas no significant differences between 20-day treated and vehicle-treated *Cdkl*5 +/− mice were observed (Figure 7B,C). No differences in total TrkB, Erk, or Akt levels were observed between vehicle-treated *Cdkl*5 +/+ and *Cdkl*5 +/− mice, or between 7- or 20-day luteolin-treated *Cdkl*5 +/− mice (Appendix A–E).

## 3. Discussion

Despite the simple genetic etiology of CDD, the brain alterations underlying its pathophysiology remain poorly understood, hindering the identification of therapeutic strategies. Inflammatory states are often associated with the main pathologies involving neurodevelopmental disorders [26], such as autistic disorders, Schizophrenia, bipolar disorder, Rett Syndrome, and Down syndrome [25,27,28,29,30,34,35,36,37]. All these conditions are linked to an inflammatory mechanism that marks the neural dysfunction. The recent demonstration of the presence of chronic microglia activation in the brain of a mouse model of CDD [52] suggests that an inflammatory state is associated with the CDD pathogenesis. Here, using a chronic luteolin treatment, we provide the first evidence that, by recovering microglia overactivation, hippocampal neurogenesis, dendritic pathology, and behavioral abnormalities were improved in heterozygous female *Cdkl*5 +/− mice, suggesting that luteolin, a natural flavonoid with anti-oxidant and anti-inflammatory properties [49,50,61,62], might have a therapeutic value in CDD.

Most CDD patients are females who are heterozygous for *CDKL*5 deficiency due to X-chromosome random inactivation (XCI). In the current study, we examined the effect that luteolin exerts in heterozygous *Cdkl*5 +/− female mice, since CDD affects mostly females, and almost all of these patients are heterozygous for a *CDKL*5 mutation [63]. Previous observations demonstrated that the mosaic loss of *Cdkl*5 in 50% of cells in the brain alters neuronal survival/maturation in a manner that is sufficient to impair behavioral performance [10,11].

Here, we found that behavioral performance in the open field and passive avoidance tests was ameliorated in luteolin-treated *Cdkl5* +/− mice. Recent findings have shown that luteolin improves hippocampal-dependent learning and memory performance in a mouse model of Down syndrome [41]. Our finding that luteolin treatment ameliorated hippocampal-dependent memory in *Cdkl*5 +/− mice supports the efficacy of luteolin in the improvement of hippocampal function in neurodevelopmental disorders. Moreover, in line with evidence that treatment with luteolin ameliorated social behaviors in a murine model of autistic behaviors [47], we found a marked improvement in motor stereotypies and hyperactivity phenotype in treated-*Cdkl*5 +/− mice, abnormalities that are linked to autistic-like behaviors, suggesting the therapeutic efficacy of luteolin in other brain regions besides the hippocampus. Since it has been well documented that chronic treatment with luteolin does not affect behavior in wild-type mice [41,64,65,66], the effect of luteolin on the *Cdkl5* −/+ mouse might be explained by the selective recovery of microglia overactivation exerted by luteolin in the *Cdkl5* KO condition. This is in line with recent evidence suggesting that microglial activation contributes to cognitive and motor impairments in mouse models of brain disorders [67,68,69].

In addition to neuroprotective effects [52], we found that treatment with luteolin increased the rate of cell proliferation in the sub-granular zone of the hippocampal dentate gyrus of *Cdkl5* +/− mice. Similarly, recent findings have demonstrated an increase in hippocampal neurogenesis in the Ts65Dn mouse, a mouse model of Down syndrome, following luteolin treatment [41] and in a rat model of Alzheimer’s Disease treated with quercetin [70], another anti-inflammatory flavonoid. Our previous evidence that a 7-day luteolin treatment, albeit sufficient to restore microglial activation to control levels [52], did not induce an increase in cell proliferation suggests that luteolin is effective at increasing the proliferation rate of granule cell precursors only after prolonged administration. Newborn granule cells (DCX + cells), however, were shown to undergo an increase in number after only 7 days of treatment in our previous study [52]; an increase that reached higher values than those of the wild-type mice after a chronic (20-day, present finding) treatment, suggesting that the pro-survival effect of luteolin on newborn neurons has a faster efficacy. The direct involvement of microglia in the survival of neuronal precursors has recently been demonstrated, with evidence showing that microglial cells residing in the subgranular zone (SGZ) and subventricular zone (SVZ), the two neurogenic niches of the rodent adult brain, play a crucial role in neuroblast survival [71,72,73], while pro-inflammatory cytokines released by activated microglia cells impair progenitor survival and differentiation [74,75]. Dendrites and spines are the main neuronal structures that receive input from other neurons, and dendritic and spine number size and morphology are some of the crucial factors for the proper functioning of the nervous system and cognitive processes. Neurodevelopmental disorders are characterized by altered neuronal maturation and by an increased number of spines with immature morphology [76]. In *Cdkl*5 +/− mice, severe defects in the cortical and hippocampal region, in terms of dendritic arborization, spine maturation, and synapse development, have been repeatedly described [7,10,11,14,15,16,17,18,19]. The current study provides novel evidence that a 20-day treatment with luteolin in adult (3-4 months) heterozygous female *Cdkl*5 +/− mice strongly improved dendritic length not only of newly formed neurons, born during the treatment period (DCX+ cells), but also of older neurons, suggesting that a proper microglial function is important both during development and maturation of neuronal cells. Interestingly, while luteolin treatment restored the total dendritic length of DCX positive cells in *Cdkl*5 +/− mice by enhancing dendritic elongation without incrementing the number of branches, in hippocampal and cortical pyramidal neurons the same improvement in dendritic length was achieved, mainly through a restoration of the number of branches. Recent studies show that microglia positively regulate developmental neurite growth through two mechanisms, direct microglia–neuron interaction and the release of soluble factors, such as IGF-1 and BDNF, known to promote neurite growth [77,78,79,80,81,82,83]. Moreover, several studies using mouse models, in which microglia cells are dysfunctional/overactive, have suggested the involvement of microglia in dendritic tree sprouting of newborn granule cells [77]. Although at the moment we have no explanation as to why treatment with luteolin differently affects elongation or number of branches in newborn and mature neurons, our results confirm the role of microglia in the modulation of neurite growth and maintenance. Interestingly, Wang and colleagues [84] found that treatment with the flavonoid 7,8-dihydroxyflavone (DHF), a small-molecule BDNF receptor agonist, enhances dendritic elongation without affecting branching of newborn granule cells in aged mice, suggesting that the effect of luteolin on dendritic arborization of DCX+ granule cells in *Cdkl*5 +/− mice may be driven by the boost in BDNF/TrkB signaling.

Along with the restoration of dendritic arbor complexity, we found that treatment with luteolin promoted the restoration of the balance between mature and immature spines in the hippocampus and cortex of *Cdkl*5 +/− mice, and that in the hippocampus of luteolin-treated *Cdkl*5 +/− mice, spine maturation correlated with an increase in the number of puncta that were immunoreactive for PSD-95, a key excitatory postsynaptic scaffold protein required for synaptic stabilization [85,86]. The contribution of microglia to the synaptic circuit remodeling and maturation through the release of soluble factors that induce changes in the molecular composition of pre- and postsynaptic compartments is well characterized [77,87,88]. On the other hand, when overactivated, microglial release of molecules, such as cytokines, chemokines, and reactive oxygen species, leads to synaptic plasticity and learning and memory deficits [89,90]. 

BDNF is one of the master regulators of dendritic development and spine density production/maturation [91,92,93]. It has been shown that luteolin treatment promotes the increase in BDNF expression in the cerebral cortex and hippocampus of mice [41,94]. We previously found that a 7-day luteolin treatment increased BDNF expression in the cerebral cortex of *Cdkl*5 KO mice [52]. Here, we show that luteolin-induced increased BDNF levels induce an increase in TrkB phosphorylation and activation of downstream pathways (Akt and Erk), important players in neuronal survival, neurogenesis, neurite outgrowth, and synaptic plasticity [59,60]. Indeed, it has been shown that persistent dual phosphorylation of Erk1/2 is important for local cytoskeletal effects and transcriptional changes that lead to dendritic remodeling in hippocampal pyramidal neurons [95]. Similarly, the Akt/mTOR signaling pathway promotes dendritic growth and branching through the upregulation of protein and lipid synthesis [96,97]. Therefore, the mechanism underlying the amelioration of dendritic pathology and behavioral defects by luteolin may be associated with the increase in BDNF. However, it is not clear why the effect of luteolin on BDNF levels, and consequently on the downstream pathways, is not retained after prolonged (20 days) luteolin treatment. Accumulating data indicate that there is a regulatory negative feedback loop between BDNF and miRNAs in the brain. Indeed, BDNF stimulates expression of neuronal miRNAs, which, in turn, act by inhibiting BDNF expression [98]. This negative feedback loop could account for the lower BDNF levels present in the cortex of 20-day luteolin-treated *Cdkl*5 +/− mice in comparison to 7-day treated mice. However, we believe that, despite the activation of BDNF-dependent pathways no longer being present in 20-day-treated *Cdkl*5 +/− mice, the molecular changes induced by the early activation of the BDNF signaling cascade are able to induce a long-lasting effect that, along with the inhibition of neuroinflammation through an attenuation of microglial activation and associated proinflammatory cytokine release [52], concur in promoting the recovery of brain function in *Cdkl*5 KO mice. This hypothesis is supported by observations showing that BDNF can induce long-lasting strengthening of synapses and neuronal survival in vivo, and this effect is dependent on transcription [99,100,101]. However, in view of the diverse actions of luteolin as an antioxidant, anticancer, and anti-inflammatory agent [61], we cannot exclude the possibility that other luteolin-activated signaling pathways contribute to the positive effect of the 20-day luteolin treatment in *Cdkl5* KO mice. 

Future studies aimed at evaluating the long-lasting effect of 20-day luteolin treatment on dendritic pathology and behavior in *Cdkl*5 +/− mice, after treatment cessation, will help to clarify whether the early activation of the BDNF signaling cascade, and consequent molecular changes, are the main mechanisms underlying the therapeutic effect of luteolin.

## 4. Materials and Methods

### 4.1. Colony and Treatments

*Cdkl*5 +/− heterozygous female mice were produced by crossing *Cdkl*5 +/− females with *Cdkl*5 +/− males and were genotyped as previously described [6,24]; age-matched wild-type *Cdkl*5 +/+ littermate controls were used for all experiments. The day of birth was designated as postnatal day (P) zero and animals with 24 h of age were considered as 1-day-old animals (P1). After weaning (P21–23), mice were housed three to five per cage on a 12-h light/dark cycle in a temperature and humidity-controlled environment, with food and water provided ad libitum. 

The animals’ health and comfort were controlled by the veterinary service. Experiments were performed in accordance with the Italian and European Community law for the use of experimental animals and were approved by Bologna University Bioethical Committee. In this study, all efforts were made to minimize animal suffering and to keep the number of animals used to a minimum. 

Experiments were carried out on a total of 100 *Cdkl*5 KO mice (*Cdkl*5 +/+ n = 32; *Cdkl*5 +/− n = 68; Appendix A). Mice were randomized as follows: *Cdkl5* +/+ and *Cdkl5* +/− females that belonged to the same litter were equally distributed within all individual groups (vehicle-treated *Cdkl5* +/+; vehicle-treated *Cdkl5* +/−; luteolin-treated *Cdkl5* +/−). All treatments were performed in the animal house at the same hour of the day. 

#### 4.1.1. Luteolin Treatment

Starting from postnatal day 90 (P90), mice were treated with a vehicle (2% DMSO in saline) or luteolin (10 mg/kg in saline; Tocris, Ellisville, MO, USA) administered through intraperitoneal injection (i.p.) daily for 7 or 20 days. The dose of luteolin was chosen based on [41]. On the day following the last treatment, mice were sacrificed for histological and Western blot analyses. 

#### 4.1.2. BrdU Treatment 

On the twentieth day of luteolin treatment, animals from all the experimental groups received a single subcutaneous injection (150 μg/g body weight) of BrdU (5-bromo-2-deoxyuridine; Sigma-Aldrich, Saint Louis, MO, USA) and were sacrificed the day after.

### 4.2. Behavioral Assays

All animal behavioral studies and analyses were performed blinded to genotype and treatment. Mice were allowed to habituate to the testing room for at least 1 h before the test, and testing was performed at the same time of day. A total of 57 animals separated into 2 independent test cohorts were used for the behavioral studies. A first test cohort consisted of 30 animals (vehicle-treated *Cdkl*5 +/+ n = 10; vehicle-treated *Cdkl*5 +/− n = 10; luteolin-treated *Cdkl*5 +/− n = 10). A second test cohort consisted of 27 animals (vehicle-treated *Cdkl*5 +/+ n = 11; vehicle-treated *Cdkl*5 +/− n = 8; luteolin-treated *Cdkl*5 +/− n = 8). Starting from the eighth day of luteolin treatment, the animals were behaviorally tested with a sequence of tests arranged to minimize the effect of one test influencing the subsequent evaluation of the next, and mice were allowed to recover for 3-6 days between different tests. 

#### 4.2.1. Hind-Limb Clasping

Animals were suspended by their tail for 2 min and hind-limb clasping time was assessed independently by two operators from video recordings. A clasping event is defined by the retraction of limbs into the body and toward the midline. 

#### 4.2.2. Open Field Test

To assess locomotion, the animals were placed in the center of a square arena (50 × 50 cm) and their behavior was monitored for 15 min using a video camera placed above the center of the arena. Distinct features of locomotor activity, including total distance traveled, average locomotion velocity, and the time spent in the center, were scored using EthoVision15XT software (Noldus, Wageningen, The Netherlands) The average locomotion velocity was calculated as the ratio between distance traveled and time. The number of stereotypical jumps (repetitive beam breaks < 1 s) was manually counted by a trained observer. Test chambers were cleaned with 70% ethanol between test subjects.

#### 4.2.3. Passive Avoidance Test 

For the passive avoidance task, a memory task that involves contributions from both the hippocampus and amygdala, we used a tilting-floor box (47 × 18 × 26 cm) divided into 2 compartments (lit and dark) by a sliding door and a control unit that incorporated a shocker (Ugo Basile, Gemonio, Italy). Upon entering the dark compartment, mice received a brief mild foot shock (0.4 mA for 3 s) and were removed from the chamber after a 15 s delay. The chambers were cleaned with 70% ethanol between the testing of one subject and another. After a 24 h retention period, mice were placed back into the illuminated compartment and the latency to reenter the dark chamber was measured up to 360 s.

### 4.3. Histological and Immunohistochemistry Procedures

Animals were anesthetized with 2% isoflurane (in pure oxygen) and sacrificed through cervical dislocation. Brains were quickly removed and cut along the midline. Left hemispheres were Golgi-stained or quickly frozen and used for Western blot analyses (see description below). 

Right hemispheres were fixed via immersion in 4% paraformaldehyde (100 mM phosphate buffer, pH 7.4) for 48 h, kept in 15–20% sucrose for an additional 24 h, and then frozen with cold ice. Hemispheres were cut with a freezing microtome into 30-μm-thick coronal sections that were serially collected and processed for immunohistochemistry procedures. One out of six sections from the hippocampal formation or somatosensory cortex were used for immunohistochemistry for doublecortin (DCX), anti-allograft inflammatory factor 1 (AIF1), Ki67, postsynaptic density protein 95 (PSD-95), or BrdU following the protocol published in [10]. Nuclei were counterstained with Hoechst 33342 (Sigma-Aldrich). The primary and secondary antibodies used are listed in Appendix A.

### 4.4. Golgi Staining

Hemispheres were Golgi-stained using the FD Rapid GolgiStain TM Kit (FD NeuroTechnologies, Columbia, MD, USA), as previously described [102]. Hemispheres were cut with a microtome into 100-µm-thick coronal sections that were directly mounted onto gelatin-coated slides and were air dried at room temperature in the dark for an additional 2–3 days. After drying, sections were rinsed with distilled water and subsequently stained in the developing solution of the kit. 

### 4.5. Image Acquisition and Measurements

Fluorescence images were taken with an Eclipse TE 2000-S microscope equipped with a DS-Qi2 digital SLR camera (Nikon Instruments Inc., Tokyo, Japan). A light microscope (Leica Mycrosystems, Shinjuku City, Tokyo, Japan), equipped with a motorized stage and focus control system, and a color digital camera (Coolsnap-Pro, Media Cybernetics, Rockville, MD, USA) were used for neuronal tracing and to take bright field images. Measurements were carried out using the Image Pro Plus software (Media Cybernetics, Silver Spring, MD, USA).

#### 4.5.1. Cell Density

The number of Ki-67+, BrdU+, and DCX+ cells were counted in the subgranular and granular zone of the dentate gyrus and expressed as number of cells/mm. The granule cell density in the dentate gyrus was evaluated as number of Hoechst+ nuclei/area and expressed as number of cells/μm^2^. 

#### 4.5.2. Morphometric Microglial Cell Analysis 

Starting from 20× magnification images of AIF-1-stained hippocampal and cortical slices, AIF-1 positive microglial cell body size was manually drawn using the Image Pro Plus measurement function and was expressed in μm^2^. The roundness index of each microglia cell was calculated as reported in [52,103].

#### 4.5.3. Measurement of the Dendritic Tree

Dendritic trees of newborn DCX+ granule neurons (15–20 per animal) and of Golgi-stained pyramidal neurons (apical and basal dendrites) of the hippocampal CA1 field and of the layers II/III of the somatosensory cortex were traced using custom-designed software for dendritic reconstruction (Immagini Computer, Milan, Italy), interfaced with Image Pro Plus, as previously described [102]. The dendritic tree was traced live, at a final magnification of 500×, by focusing on the depth of the section. The operator starts with branches that emerge from the cell soma and after having drawn the first parent branch, proceeds with all the daughter branches of the next order in a centrifugal direction. At the end of tracing, the program reconstructs the number and length of individual branches, the mean length of branches of each order, and the total dendritic length.

#### 4.5.4. Dendritic Spine Number and Morphology

In Golgi-stained sections, dendritic spines of hippocampal and cortical pyramidal neurons were visualized with a 100× oil immersion objective lens. Based on their morphology, dendritic spines can be divided into two different categories that reflect their state of maturation, immature spines and mature spines. The number of spines that belong to the two different groups (immature spines: filopodium-like, thin- and stubby-shaped; mature spines: mushroom- and cup-shaped) was counted and expressed as a percentage. About 200–250 spines from 25 to 30 dendrites, derived from 10 to 20 neurons, were analyzed per condition.

#### 4.5.5. Quantification of PSD-95 Immunoreactive Puncta

Images from the hippocampal CA1 layer were acquired using a LEICA TCS SL confocal microscope (LEITZ; Leica Microsystems, Wetzlar, Germany; objective 63×, NA 1.32; zoom factor = 8). Three to four sections per animal were analyzed and in each section, three images from the regions of interest were captured and the density of individual puncta exhibiting PSD-95 immunoreactivity was evaluated as previously described [19,52]; the number of PSD-95 immunoreactive puncta was expressed per μm^2^.

### 4.6. Western Blotting

For the preparation of total cell extracts, tissue samples were homogenized in RIPA buffer and quantified using the Bradford method, as previously described [52]. Equivalent amounts (50 μg) of protein were subjected to electrophoresis on a 4–12% Mini- PROTEAN^®^ TGX™ Gel (Bio-Rad, Hercules, CA, USA) and transferred to a Hybond ECL nitrocellulose membrane (GE Healthcare Bio-Science, Piscataway, NJ, USA). The primary and secondary antibodies used are listed in Appendix A. The densitometric analysis of digitized Western blot images was performed using Chemidoc XRS Imaging Systems and the Image LabTM Software (Bio-Rad); this software automatically highlights any saturated pixels of the Western blot images in red. Images acquired with exposition times that generated protein signals out of a linear range were not considered for the quantification.

### 4.7. Statistical Analysis

Statistical analysis was performed using GraphPad Prism (version 7). Values are expressed as means ± standard error (SEM). The significance of results was obtained using two-tailed Student’s *t*-test or one-way or two-way ANOVA, followed by Fisher’s LSD post hoc test, as specified in the figure legends. A probability level of *p* < 0.05 was considered to be statistically significant. The confidence level was taken as 95%.

## 5. Conclusions

Although pharmacotherapy has shown promise in the CDD mouse model [14,17,18,19,20,21,104,105], safe and effective clinical treatments are still far off. The present data provide a clear picture of how deeply neuroinflammatory processes contribute to the brain alterations in *Cdkl*5 mice and how effective an anti-inflammatory treatment with luteolin is in recovering the brain dysfunctions. Luteolin, a natural flavonoid that is able to freely penetrate the blood–brain barrier due to its lipophilicity [106], is safe and has beneficial effects in several mouse models of brain disorders [49]. Moreover, in clinical trials with dietary supplements that contain luteolin, treatment has been found to be safe and, more importantly, effective in promoting an improvement in autism spectrum disorder behavioral symptoms in autistic children [107,108,109]. Therefore, luteolin could represent a valid and beneficial adjuvant therapeutic option to ameliorate brain functionality in patients affected by CDD. 

## Figures and Tables

**Figure 1 ijms-23-08719-f001:**
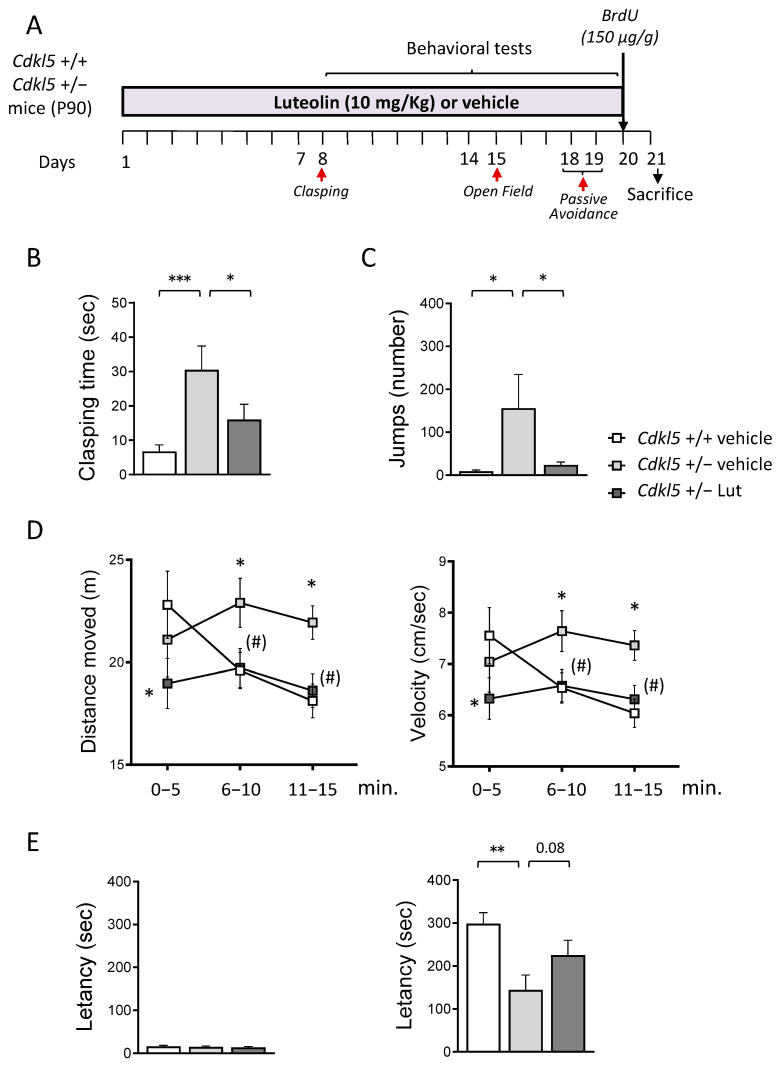
Effect of luteolin treatment on behavior of *Cdkl*5 +/− mice. (**A**) Experimental plan. Starting from postnatal day 90 (P90), *Cdkl*5 +/− female mice were treated with vehicle or luteolin (i.p. 10 mg/kg) for 20 days. Behavioral tests were performed during the last two weeks of treatment. Red arrows indicate the day on which the behavioral tests were performed. The day before sacrificed mice received a single subcutaneous injection of bromodeoxyuridine (BrdU 150 µg/g). (**B**) Total amount of time spent hind-limb clasping during a 2-min interval in vehicle-treated *Cdkl*5 +/+ (n = 21) and *Cdkl*5 +/− (n = 18) mice and luteolin-treated *Cdkl*5 +/− (n = 18) mice. (**C**) Number of stereotypic jumps (repetitive beam breaks < 1 s) in the corners of the open field arena during the 15-min trial in vehicle-treated *Cdkl*5 +/+ (n = 19) and *Cdkl*5 +/− (n = 18) mice and luteolin-treated *Cdkl*5 +/− (n = 17) mice. (**D**) Locomotor activity measured as average locomotion velocity (right graph) and total distance travelled (left graph) during a 15-min open field test in vehicle-treated *Cdkl*5 +/+ (n = 19) and *Cdkl*5 +/− (n = 18) mice and luteolin-treated *Cdkl*5 +/− (n = 17) mice. (**E**) Passive avoidance test on vehicle-treated *Cdkl*5 +/+ (n = 21) and *Cdkl*5 +/− (n = 18) mice and luteolin treated *Cdkl*5 +/− (n = 17) mice. Graphs show the latency to enter the dark compartment on the 1st day (on the left) and on the 2nd day (on the right) of the behavioral procedure. Values represent mean ± SEM. * *p* < 0.05, ** *p* < 0.01, *** *p* < 0.001 as compared to the vehicle-treated *Cdkl*5 +/+ condition; (#) *p* < 0.065 as compared to the vehicle-treated *Cdkl*5 +/− condition. Fisher’s LSD test after one-way ANOVA for data set in (**B**,**C**,**E**) and after two-way ANOVA for data set in (**D**).

**Figure 2 ijms-23-08719-f002:**
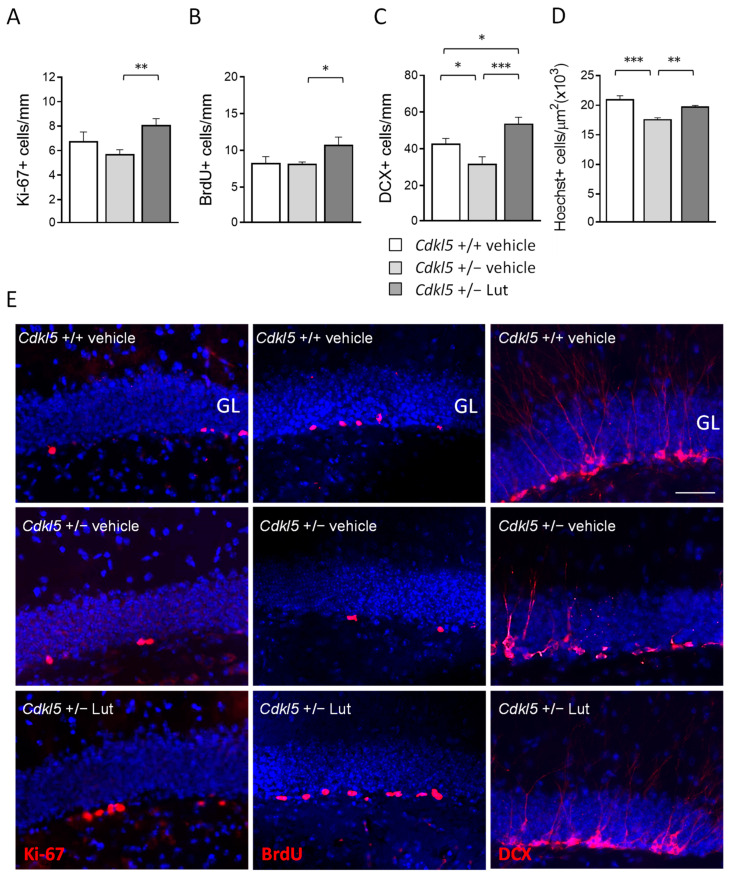
Effect of luteolin treatment on hippocampal neurogenesis of *Cdkl*5 +/− mice. (**A**) Number of Ki-67-positive cells in the subgranular zone (SGZ) of the dentate gyrus (DG) of vehicle-treated *Cdkl*5 +/+ (n = 8) and *Cdkl*5 +/− (n = 9) mice and luteolin-treated *Cdkl*5 +/− (n = 8) mice. (**B**) Number of BrdU-positive cells in the SGZ of vehicle-treated *Cdkl*5 +/+ (n = 4) and *Cdkl*5 +/− (n = 4) mice and luteolin-treated *Cdkl*5 +/− (n = 4) mice. (**C**) Quantification of DCX-positive cells in the granular layer (GL) of the DG of vehicle-treated *Cdkl*5 +/+ (n = 7) and *Cdkl*5 +/− (n = 7) mice and luteolin-treated *Cdkl*5 +/− (n = 7) mice. (**D**) Quantification of Hoechst-positive cells in the GL of the DG of vehicle-treated *Cdkl*5 +/+ (n = 4) and *Cdkl*5 +/− (n = 4) mice and luteolin-treated *Cdkl*5 +/− (n = 4) mice. (**E**) Examples of sections processed for fluorescent immunostaining for Ki-67 (left), BrdU (middle), and DCX (right) from the DG of an animal from each experimental condition. Scale bar = 50 μm. GL = granule cell layer. Values are represented as means ± SEM. * *p* < 0.05, ** *p* < 0.01, *** *p* < 0.001. Fisher’s LSD test after one-way ANOVA.

**Figure 3 ijms-23-08719-f003:**
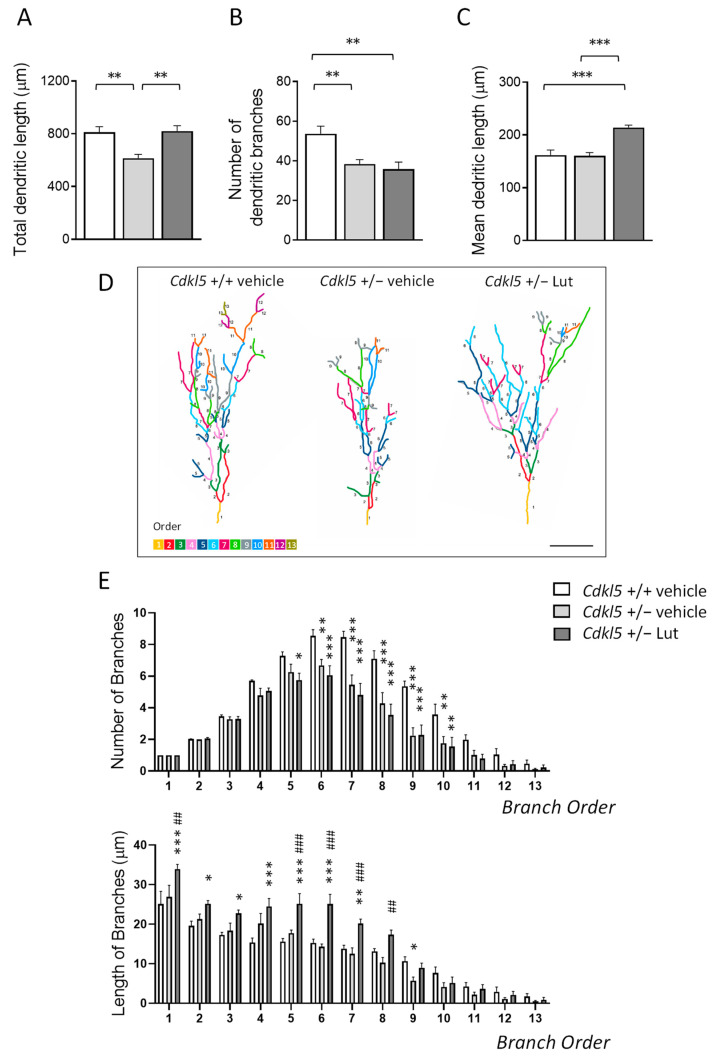
Effect of luteolin treatment on dendritic development of newborn granule cells in the DG of *Cdkl*5 +/− mice. (**A**–**C**) Total dendritic length (**A**), mean number of dendritic branches (**B**) and mean dendritic length (**C**) of DCX-positive cells in the dentate gyrus (DG) of vehicle-treated *Cdkl*5 +/+ (n = 5) and *Cdkl*5 +/− (n = 5) mice and luteolin-treated *Cdkl*5 +/− (n = 6) mice. (**D**) Examples of the reconstructed dendritic tree of DCX+ granule cell of an animal from each experimental condition. Scale bar = 40 μm. (**E**) Quantification of the mean number (upper histogram) and mean length (lower histogram) of branches of different orders in DCX+ granule cells of *Cdkl*5 KO mice treated as in (**A**). Dendrites were traced in a centrifugal direction. Numbers indicate the different dendritic branch orders. Values are represented as means ± SEM. * *p* < 0.05, ** *p* < 0.01, *** *p* < 0.001, as compared to the vehicle-treated *Cdkl*5 +/+ condition; ## *p* < 0.01; ### *p* < 0.001, as compared to the vehicle-treated *Cdkl*5 +/− condition. Fisher’s LSD test after one-way ANOVA for data set in (**A**–**C**), and after two-way ANOVA for data set in (**E**).

**Figure 4 ijms-23-08719-f004:**
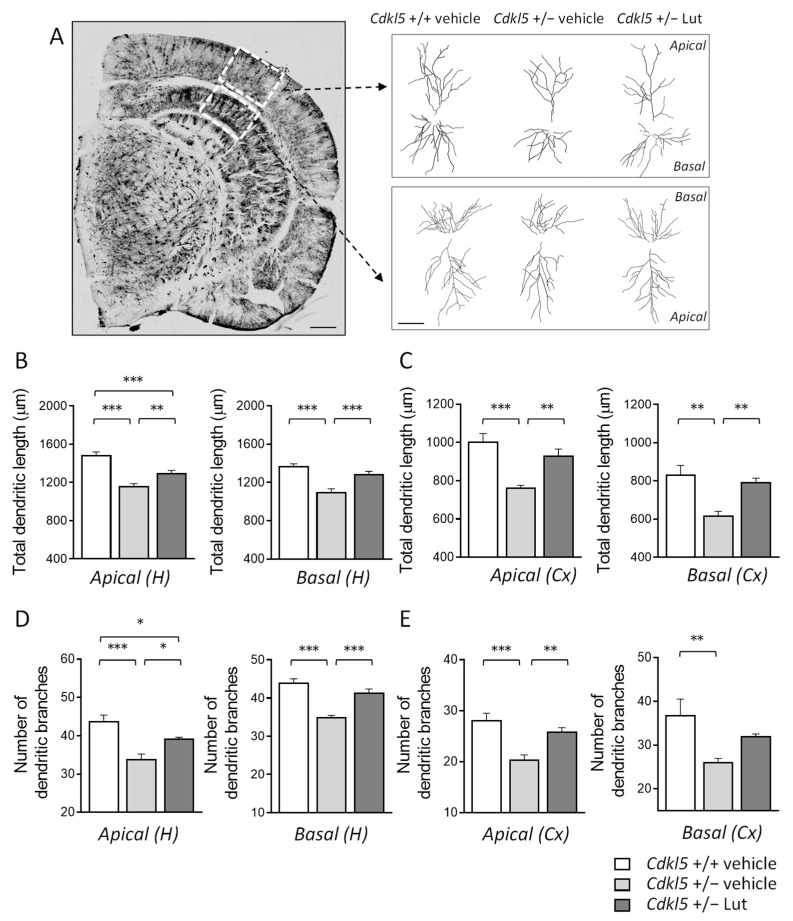
Effect of luteolin treatment on dendritic architecture of hippocampal and cortical neurons of *Cdkl*5 +/− mice. (**A**) Representative image of a Golgi-stained section (panel on the left; scale bar = 500 μm), showing the portion of hippocampal and cortical regions where the dendritic arbor of pyramidal neurons was traced (areas enclosed in the dashed square). On the right, examples of the reconstructed apical and basal dendritic tree of Golgi-stained cortical (upper) and CA1 (lower) pyramidal neurons of one animal from each experimental group. Scale bar = 40 μm. (**B**,**C**) Total dendritic length of apical (on the left) and basal (on the right) dendrites of Golgi-stained pyramidal neurons in the CA1 field (**B**) and cortex (**C**) of vehicle-treated *Cdkl*5 +/+ (n = 4) and *Cdkl*5 +/− (n = 4) mice and luteolin-treated *Cdkl*5 +/− (n = 4) mice. (**D**,**E**) Number of dendritic branches of apical (on the left) and basal (on the right) dendrites of Golgi-stained pyramidal neurons in the CA1 field (**D**) and cortex (**E**) of mice as in (**B**). H = hippocampus; CX = cortex. Values are represented as means ± SEM. * *p* < 0.05, ** *p* < 0.01, *** *p* < 0.001. Fisher’s LSD test after one-way ANOVA.

**Figure 5 ijms-23-08719-f005:**
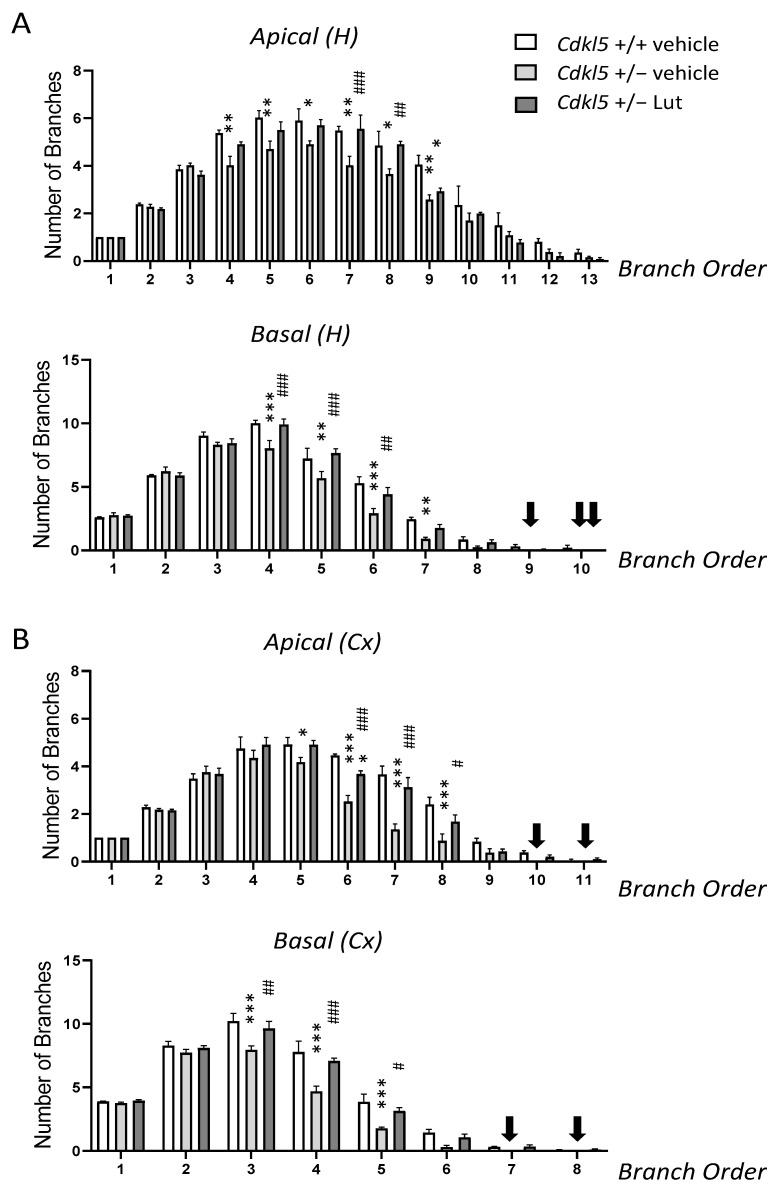
Effect of luteolin treatment on dendritic arborization complexity of hippocampal and cortical neurons in *Cdkl*5 +/− mice. (**A**,**B**) Quantification of the number of branches of different orders of apical dendrites (upper panel) and basal dendrites (lower panel) in Golgi-stained hippocampal (**A**) and cortical (**B**) pyramidal neurons of vehicle-treated *Cdkl*5 +/+ (n = 4) and *Cdkl*5 +/− (n = 4) mice and luteolin-treated *Cdkl*5 +/− (n = 4) mice. Dendrites were traced in a centrifugal direction. Numbers indicate the different dendritic branch orders. Black arrows indicate the lack of branches of that order. H = hippocampus; CX = cortex. Values are represented as means ± SEM. * *p* < 0.05, ** *p* < 0.01, *** *p* < 0.001, as compared to the vehicle-treated *Cdkl*5 +/+ condition; # *p* < 0.05, ## *p* < 0.01, ### *p* < 0.001, as compared to the vehicle-treated *Cdkl*5 +/− condition. Fisher’s LSD test after two-way ANOVA.

**Figure 6 ijms-23-08719-f006:**
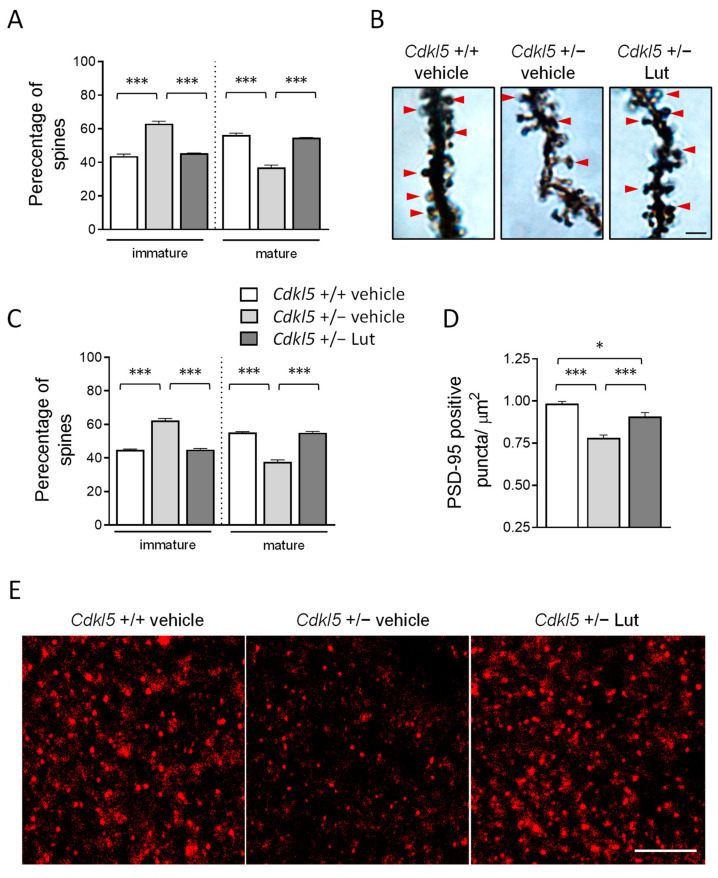
Effect of luteolin treatment on neuronal connectivity in *Cdkl*5 +/− mice. (**A**,**C**) Percentage of immature and mature dendritic spines in relation to the total number of protrusions in CA1 (**A**) and cortical (**C**) pyramidal neurons of vehicle-treated *Cdkl*5 +/+ (n = 4) and *Cdkl*5 +/− (n = 4) mice and luteolin-treated *Cdkl*5 +/− (n = 4) mice. (**B**) Examples of Golgi-stained hippocampal pyramidal neurons of one animal from each experimental group; red arrows represent mature spines. Scale bar = 2 μm. (**D**) Number of fluorescent puncta per μm^2^, exhibiting PSD-95 immunoreactivity in the CA1 layer of the hippocampus of vehicle-treated *Cdkl*5 +/+ (n = 4) and *Cdkl*5 +/− (n = 4) mice and luteolin-treated *Cdkl*5 +/− (n = 4) mice. (**E**) Representative fluorescence image of PSD-95 immunoreactive puncta in the hippocampus of one animal from each experimental group. Scale bar = 6 μm. Values are represented as means ± SEM. * *p* < 0.05, *** *p* < 0.001. Fisher’s LSD test after two-way ANOVA.

**Figure 7 ijms-23-08719-f007:**
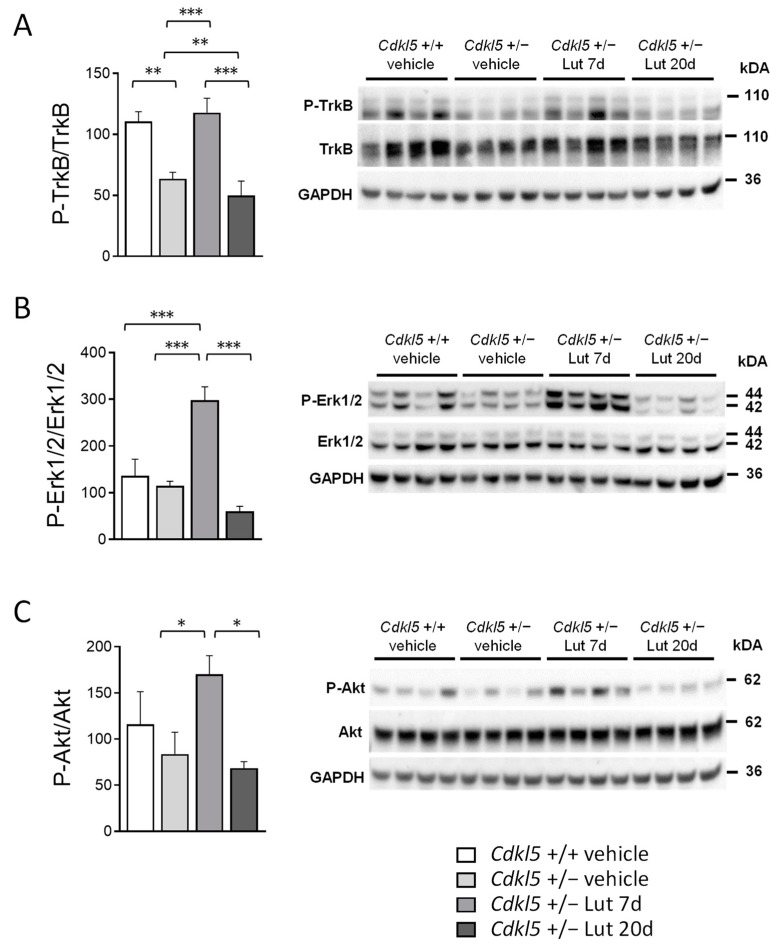
Effect of luteolin treatment on BDNF/TrkB signaling pathway activation in the cortex of *Cdkl*5 +/− mice. (**A**) Western blot analysis of P-TrkB levels in somatosensory cortex homogenates from vehicle-treated *Cdkl*5 +/+ (n = 4) and *Cdkl*5 +/− (n = 4) mice, 7-day luteolin treated *Cdkl*5 +/− (Lut 7d, n = 4) and 20-day luteolin treated *Cdkl*5 +/− (Lut 20d, n = 4) mice. The histogram on the left shows P-TrkB protein levels normalized to corresponding total TrkB protein levels. Examples of immunoblot for P-TrkB on the right. (**B**) Histogram on the left shows protein levels of phosphorylated Erk1/2 normalized to the respective total form in cortex homogenates from mice treated as in (**A**). Examples of immunoblot for P-Erk1/2 on the right. (**C**) Histogram on the left shows protein levels of phosphorylated Akt normalized to the respective total form in cortex homogenates from mice treated as in (**A**). Examples of immunoblot for P-Akt on the right. Data are expressed as a percentage of vehicle-treated *Cdkl*5 +/+ mice. Values represent mean ± SEM. * *p* < 0.05, ** *p* < 0.01, *** *p* < 0.001. Fisher’s LSD test after two-way ANOVA.

## Data Availability

The datasets analyzed during the current study are available from the corresponding author upon reasonable request.

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
