# Peer review of "Luteolin Treatment Ameliorates Brain Development and Behavioral Performance in a Mouse Model of CDKL5 Deficiency Disorder"

_ijms, 2022, doi:10.3390/ijms23158719_

Round 1
Reviewer 1 Report
This manuscript entitled "Luteolin treatment ameliorates brain development and behavioral performance in a mouse model of CDKL5 deficiency disorder " by Tassinari et al. is interesting and a good contribution to the field of herbal medicine and neuroinflammation. The manuscript needs some revisions before it can be recommended for publication.
Abstract
-The design of study can be defined with more details.
-The main results of study can be mentioned.
- The keywords should be checked by MeSH tools.
Introduction
-The authors mentioned briefly about the Luteolin efficiency and its pharmacological effects with focus on anti-inflammatory properties. Authors need to elaborate effectively.
You can use the following review:
- The Flavone Luteolin Improves Central Nervous System Disorders by Different Mechanisms: A Review
- It is logic to explain about the disease first, then the pathophysiology of CDD can be discussed and the role of microglia can be highlighted in this part.”
Result
- The explanation of each figure should be done in one paragraph under the related subtitle. Authors explained the anti-inflammatory effect of luteolin in 3.1. part.
-The figures and their legends should be placed after their definitions.
-There is no need to discuss the results in this part. Consistently with previous studies [34,50], ….
-In Figure 3E and figure 5, the different orders should be defined.
Discussion
-The study should be explained in one paragraph first and the model should be discussed and the main results associated with the model group should be compared with other studies.
-To be logic, the results are important to be compared with other studies.
-Before explaining about other studies, the main results of this study should be explained and then the findings should be compared with other studies and possible mechanisms can be added.
Reference:
-Authors need to use most recently published articles and references should be updated.
Reviewer 2 Report
In this manuscript, Tassinari et al. investigated the effect of luteolin on behavioral, structural and signaling deficits in Cdkl5 KO mice, which is an animal model of CDKL5 deficiency disorder (CDD). They found that luteolin treatment for 7 days, but not for 20 days, enhanced p-Erk and p-Akt signaling and rescued pTrkB signaling deficits caused by Cdkl5 mutation. They also found that luteolin treatment for 20 days improved neurogenesis, neuronal survival, and spine maturation, and recovered microglia activation in Cdkl5 KO mice. Possibly due to its rescue effect on spine structure, luteolin treatment for 20 days also improved behavioral deficits in Cdkl5 KO mice. The results are consistent with their recent publication “Inhibition of microglia overactivation restores neuronal survival in a mouse model of CDKL5 deficiency disorder” which shows that luteolin inhibits microglia overactivation and restores neuronal survival and may provide insight into the relationship between neuroinflammation and cognitive impairment.
Questions and comments:
1. The manuscript mentioned that “twenty-day treatment with luteolin fully recovered these microglia alterations (Figure S2A,B) both in the hippocampus and somatosensory cortex, suggesting that inhibition of microglia overactivation underlies, at least in part, the behavioral deficits in Cdkl5 KO mice”. To conclude the inhibition of microglia overactivation is a reason for behavioral rescue, the authors need to cite references to show that microglia activation contributes to deficits specifically in the three behavioral tests (clasping, open field, and inhibitory avoidance).
2. Figure 2 assesses the effect of luteolin treatment on neuronal survival with DCX staining, and Cdkl5+/- mice show a lower number of DCX+ cells compared to WT, suggesting lower neuronal survival in DG of Cdkl5 +/- mice. Do Cdkl5 +/- mice have less neuronal density in hippocampus (specifically in DG)?
3. In the discussion, although it is mentioned that “it has been well documented that chronic treatment with Luteolin does not affect behavior in wild-type mice”, it is better to have a WT plus luteolin group, at least in behavior, spine maturation and TrkB signaling experiments, to help to examine the effects of luteolin on behavior, spine maturation, and TrkB signaling.
4. In Figure 7, only Cdkl5+/- Lut 7d group, but not Cdkl5+/- Lut 20d group, enhanced p-TrkB, pErk and pAkt. If the signaling pathways are important for dendritic structural and behavioral changes in Cdkl5 KO mice, how to explain the effect of luteolin treatment for 20 days on dendritic structure and mouse behavior? If waited for some weeks after stopping the 20-day luteolin treatment, will the dendritic and behavioral deficits come back in Cdkl5 KO mice?
Round 2
Reviewer 1 Report
Accept in present form
Reviewer 2 Report
Thank the authors for addressing my questions. I understand that adding a WT plus luteolin group may take long time to run the experiment, and I am satisfied with all the responses except the last one. I think the most proper way to answer the question "how to explain the effect of luteolin treatment for 20 days on dendritic structure and mouse behavior" is to wait for some weeks after stopping the 20-day luteolin treatment, and to measure whether the rescue for dendritic and behavioral deficits still exist. The authors cited a reference to show that BDNF has long-lasting effect after treatment cessation, but I think the study (Vink et al. 2021) does not address the question here. The preliminary result with AIF+ cell body size measurement helps to show long term effect on microglia activation after ceasing luteolin treatment, but doesn't provide information for dendritic or especially the behavioral phenotypes.
Author Response
We apologize to the Reviewer for having misinterpreted the question; we thought that the Reviewer had asked two separate questions. We agree that the best way to explain the effect/mechanisms of action of luteolin treatment for 20 days on dendritic structure and behavior in Cdkl5 KO mice is to evaluate whether or not the effect lasts after treatment cessation. We have now added this comment to the Discussion section: “Future studies aimed at evaluating the long-lasting effect of 20-day luteolin treatment on dendritic pathology and behavior in Cdkl5 +/- mice, after treatment cessation, will help to clarify whether the early activation of the BDNF signaling cascade, and consequent molecular changes, are the main mechanisms underlying the therapeutic effect of luteolin” (page 16, lines 21-24).
Round 3
Reviewer 2 Report
Thank the authors for their response.
Although I am still confused with the results in Fig7 that only Cdkl5+/- Lut 7d group but not Cdkl5+/- Lut 20d group enhanced the signaling, and I believe that the best way to test the hypothesis that “molecular changes induced by the early activation of the BDNF signaling cascade are able to induce a long-lasting effect (on behavior)” is to stop the luteolin treatment for some time (e.g., 2 weeks) and run the behavioral tests to examine whether the rescue effect still exist (otherwise the results in Fig7 don’t help to explain the mechanism of luteolin on dendritic structure and behavior), I accept the revised discussion to address this question in future studies.